# A New Therapeutic Trend: Natural Medicine for Ameliorating Ischemic Stroke via PI3K/Akt Signaling Pathway

**DOI:** 10.3390/molecules27227963

**Published:** 2022-11-17

**Authors:** Xian Liu, Xinyu Xiao, Xue Han, Lan Yao, Wei Lan

**Affiliations:** 1College of Traditional Chinese Medicine, Xinjiang Medical University, Urumqi 830017, China; 2Dermatological Department, Hospital of Chengdu University of Traditional Chinese Medicine, Chengdu 610015, China

**Keywords:** natural medicine, ischemic stroke, PI3K/Akt signaling pathway, treatment, new trend

## Abstract

Ischemic stroke (IS) is an acute cerebrovascular disease caused by sudden arterial occlusion, which is characterized by a high morbidity, mortality, and disability rate. It is one of the most important causes of nervous system morbidity and mortality in the world. In recent years, the search for new medicine for the treatment of IS has become an attractive research focus. Due to the extremely limited time window of traditional medicine treatment, some side effects may occur, and accompanied by the occurrence of adverse reactions, the frequency of exploration with natural medicine is significantly increased. Phosphatidylinositol-3-kinase/Protein kinase B (PI3K/Akt) signaling pathway is a classical pathway for cell metabolism, growth, apoptosis, and other physiological activities. There is considerable research on medicine that treats various diseases through this pathway. This review focuses on how natural medicines (including herbs and insects) regulate important pathophysiological processes such as inflammation, oxidative stress, apoptosis, and autophagy through the PI3K/Akt signaling pathway, and the role it plays in improving IS. We found that many kinds of herbal medicine and insect medicine can alleviate the damage caused by IS through the PI3K/Akt signaling pathway. Moreover, the prescription after their combination can also achieve certain results. Therefore, this review provides a new candidate category for medicine development in the treatment of IS.

## 1. Introduction

Ischemic stroke (IS) is an important cause of neurological morbidity, it can lead to high mortality and disability rates. Among all cerebrovascular diseases, IS has the highest incidence [1]. It accounts for more than 80% of all stroke types [2]. According to incomplete statistics, about 14 million people suffered from IS every year [3]. The main causes of IS include cardiogenic cerebral infarction, rupture of cerebrovascular atherosclerotic plaques, atherosclerotic plaques, and lacunar infarction caused by small vessel lesions [4]. With the aging of the population, the incidence of IS has increased dramatically. Many patients with basic diseases are more likely to suffer from IS, such as diabetes, hypertension, atrial fibrillation, and arterial disease. It was also found that hospitalized patients with Corona Virus Disease 2019 (COVID-19) had a risk of IS [5]. Therefore, the effective treatment of IS is of great significance to society as a whole. The treatment of IS includes surgical intervention and drug intervention. However, the cost and risk of surgical intervention is high. Therefore, drug intervention has become the most commonly used clinical measures, such as anti-platelet drugs and thrombolytic drugs [6]. However, the therapeutic time window of this kind of drug is extremely limited, with the risk of bleeding or even more serious consequences. This makes the search for new and effective natural medicines the key to the treatment of IS. Natural medicine, including herbal and animal medicine, is collectively referred to as traditional Chinese medicine in China [7]. It is widely used in the treatment of various diseases. Traditional Chinese Medicine (TCM) has been used effectively in the treatment of stroke for more than 2000 years in China. Over the years, this field has gleaned extensive clinical treatment experience [8]. Modern medicine has also found that natural medicine is effective in the treatment of IS, and it plays a significant preventive role in the early stage of IS [9]. In particular, natural medicine can treat IS through different signaling pathways, including Janus Kinase/Signal Transducer and Activator of Transcription (JAK/STAT), Nuclear Factor kappa-B (NF-κB), Mitogen-Activated Protein Kinase (MAPK), Nuclear Factor erythroid 2-Related Factor 2 (Nrf2), and PI3K/Akt signaling pathways [10]. Among them, the PI3K/Akt signaling pathway is involved in many diseases and is presently being widely studied.

The PI3K/Akt signaling pathway is an indispensable signal mechanism in the biological process of mammals, which is involved in signal transduction. Such as cell development, differentiation, cell survival, protein synthesis, and metabolism [11]. PI3K is a dimer composed of regulatory subunit P85 and catalytic subunit p110, which can bind to growth Akt allosteric [8]. Akt, as the main molecule downstream of PI3K signaling pathway, includes three subtypes: Akt 1, Akt 2, and Akt 3. They are encoded by Akt α, Akt β, and Akt γ, respectively [12]. Activation of Akt can further regulate its downstream substrate and cause a series of cascade reactions, thus acting in apoptosis and cell cycle regulation. Abnormal activity of PI3K/Akt signaling pathway may cause a variety of diseases, such as diabetes, neurodegenerative diseases, stroke, and cancer [13]. It has been confirmed that the PI3K/Akt signaling pathway plays an important role in oxidative stress and inflammation in the course of IS [14]. In this paper, we reviewed the evidence for natural medicine treatment of IS through the PI3K/Akt signaling pathway in recent years. We also explained its mechanism to provide new treatment ideas for the clinical treatment of IS.

## 2. Correlation between PI3K/Akt Signaling Pathway and Ischemic Stroke

### 2.1. Structural Characteristics and Activation of PI3K and Akt

According to the structural and functional differences. PI3K can be divided into I, II, and III subtypes. The subtypes I and II participate in cellular signal transduction, while subtypes III participate in membrane transport. Indeed, type I PI3K is the most widely studied and most closely related to IS at present. It encompasses regulatory subunits (p85α, p85β, and p85γ) and catalytic subunits (p110α, p110β, p110δ, and p110γ), which can be directly activated by cell surface receptors [15]. IA PI3K is activated by receptor tyrosine kinase (RTK), G protein coupled receptor, and small G protein RAS. IB PI3K is activated only by G protein coupled receptors [16]. Subsequently, the p85 subunit binds to the tyrosine receptor on the cell membrane, further activating PI3K and phosphorylation. Then, the p110 catalytic subunit is activated to catalyze the conversion of phosphatidylinositol diphosphate (PIP2) to phosphatidylinositol triphosphate (PIP3) [17]. PIP3 continues to phosphorylate and transmits its activation signal to Akt [18].

Akt is also known as protein kinase B (PKB). There are three subtypes: Akt1, Akt2, and Akt3 [19]. Different Akt subtypes are expressed in specific tissues. It indicates that Akt plays a key role in maintaining the physiological function of different tissues or organs [20]. Akt contains a special PH structural domain. If this domain has a defection, the activity of Akt also reduces or is even inactivated. After PI3K activation, the PH structural domain of Akt binds to the PI3K product PIP3 in order to promote the migration of Akt from intracellular to thin cell membrane and change the conformation. Then, the serine/threonine protein is phosphorylated, resulting in the complete activation of Akt [21]. Activated Akt further regulates its downstream substrate, causing a series of cascade reactions, and acts in apoptosis and cell cycle regulation [22] (Figure 1).

### 2.2. Ischemic Stroke Activates PI3k/Akt Signaling Pathway, Link to the Relevant Cascade Reaction

#### 2.2.1. The Pathological Mechanism of Ischemic Stroke

Ischemic stroke is a complex pathophysiological process. After cerebral ischemia, irreversible neuronal necrosis occurs in the central region due to the interruption of blood supply and energy depletion. For the ischemic penumbra, the disturbance of glucose energy metabolism leads to the decrease in Na^+^/K^+^-ATP enzyme activity and the imbalance of ion balance causes a large amount of Ca^2+^ influx, leading to the excessive release of glutamate and excitatory toxicity [23]. Glutamate further binds to the receptor and once again promotes the influx of Ca^2+^, leading to mitochondrial dysfunction and apoptosis. Meanwhile, after cerebral infarction, macrophages and microglia are activated to release vasoactive mediators and pro-inflammatory cytokines, which promote the infiltration of more leukocytes and cause neuroinflammation [24]. Inflammatory cells can also produce reactive oxygen species (ROS) and reactive nitrogen (RNS). This in turn reactivates inflammatory cells and a vicious cycle begins [25]. With the recovery of blood flow and oxygen supply after long-term infarction, ROS and RNS also increase oxidative stress. This aggravates the damage to the blood–brain barrier (BBB) and further leads to extravasation of plasma components, continued enhancement of inflammatory response, and obvious activation of autophagy-related signaling pathways [26] (Figure 2).

#### 2.2.2. Neuroinflammation Caused by Ischemic Stroke with PI3K/Akt Signaling Pathway

Cerebral ischemia can disrupt the homeostasis between pro- and anti-inflammatory responses. Clinical studies have shown that inhibition of inflammatory response can reduce brain injury and improve neurological function. However, it has also been found that the inhibition of the inflammatory response may also worsen brain repair and long-term functional recovery after IS [27]. On the one hand, inflammatory reaction in cerebral ischemic area can cause excessive immune reaction and release inflammatory chemokine such as TNF-α and IL-6 into the blood, causing extensive damage to nerve cells [28]. On the other hand, a hypoxic environment promotes the activation of microglia and the release of superoxide, which can activate and recruit neutrophils. Eventually, the blood–brain barrier is destroyed [29]. Additionally, NF-κB is a key regulator in the process of inflammation, which regulates the active expression of pro-inflammatory genes [30]. Studies have shown that the activation of PI3K/Akt signaling pathway can inhibit the expression of pro-inflammatory factors induced by NF-κB, and transform M1 microglia into M2 microglia to reduce the inflammatory response [31,32]. Additionally, the activation of this pathway can also up-regulate the expression levels of interleukin-1RA (IL-1RA), interleukin-10 (IL-10), and interferon-β (IFN-β) and decrease the expression levels of proinflammatory cytokines, including IL-1, TNF-α, IL-6, interleukin-8 (IL-8), and chemokine (C-X-C motif) ligand 1 protein (CXCL1) [15]. 

Moreover, accumulating data indicate the role of PI3K/Akt signaling pathway in inflammation in IS. Many drugs have been found to inhibit inflammation of IS through the signaling PI3K/Akt signaling pathway. It has been confirmed that inhibition of chemokine CXCL8 may promote neuroglial activation and inhibit neuroinflammatory by regulating the PI3K/Akt/NF-κB signaling pathway in mice with IS [33]. Similarly, vinpocetine reduces inflammation induced by cerebral ischemia/reperfusion (I/R) injury in brain tissues through the PI3K/Akt signaling pathway [34]. Additionally, Bisperoxovanadium (BPV) protects nerves by up-regulating PI3K and Akt to inhibit the inflammatory response caused by cerebral I/R injury in rats with IS [35]. Fraxetin has a suppression effect on microglia-mediated neuroinflammation, and this effect is associated with the PI3K/Akt signaling pathway. Furthermore, the administration of fraxetin attenuated brain injury and behavioral deficits after IS [36]. In general, all these data suggest that alleviating brain inflammation through the PI3K/Akt signaling pathway is an effective way to improve the damage caused by IS.

#### 2.2.3. Apoptosis Caused by Ischemic Stroke with PI3K/Akt Signaling Pathway

Apoptosis begins several hours after cerebral ischemia attack, which mainly appeared in ischemic penumbra. Bcl-2 family genes and cysteine protease (Caspase) family genes participate in ischemic neuronal apoptosis, which is a mitochondrial centered apoptosis process [37]. Under the stress of cerebral ischemia, the expression of Bcl2 decreases and the expression of Bax increases. It leads to caspase activation and an apoptosis-dependent cascade. Activated caspase (especially Caspase-1 and Caspase-3) can modify proteins and play a key role in the early stage of ischemia-mediated apoptosis [38]. Apoptosis has an inseparable relationship with the PI3K/Akt signaling pathway. 

It has been found that there are a series of phosphorylation cascades in the PI3K/Akt signaling pathway, which can delay cell apoptosis [39], such as insulin-like growth factor 1 (IGF-1) up-regulating the yes-associated protein/transcription activator with PDZ binding motif (YAP/TAZ) by activating PI3K/Akt signal cascade, so as to reduce the apoptosis of nerve cells in IS [40]. Additionally, it has also been found that electroacupuncture (EA) preconditioning can inhibit the expression of pro-apoptotic genes and proteins in the middle cerebral artery (MCAO) model rats. The mechanism of reducing apoptosis can be achieved through the PI3K/Akt signaling pathway [41]. Moreover, resveratrol can significantly improve the neural function and attenuate neuronal apoptosis in IS rats through activating the PI3K/Akt signaling pathway [42]. Thus, it can be seen that the activation of the PI3K/Akt signaling pathway can alleviate the brain injury caused by IS by delaying apoptosis.

#### 2.2.4. Oxidative Stress Caused by Ischemic Stroke with PI3K/Akt Signaling Pathway

ROS is a highly active substance formed by enzymatic or non-enzymatic reaction in mammalian cells, which is involved in many pathological processes during I/R [43]. Oxidative stress occurs when excessive production of ROS and/or ROS degradation is impaired, which is one of the important pathological mechanisms of IS [44]. The brain accounts for only 2% of the body weight, but oxygen consumption is close to 20%. This is why the brain produces more free radicals than other organs. Therefore, the exploration of oxidative stress reaction can better provide new ideas for the treatment of IS. After IS, the PI3K/Akt signaling pathway is activated. The downstream glycogen synthase kinase-3β (GSK-3β) is phosphorylated, resulting in the loss of the ability to activate NF-κB so as to reduce inflammation-mediated nerve cell injury [45]. At the same time, activated Akt further acts on Nrf2, a key regulator of oxidative stress. It prevents kelch-like epichlorohydrin-associated protein-1 (Keap1) from coupling with it and promotes the binding of the antioxidant response element (ARE) to it. Finally, it enhances the expression of antioxidant proteins [46]. 

Emerging evidence has shown that the activation of the PI3K/Akt signaling pathway can effectively reduce oxidative stress induced by IS [47]. Studies have shown that basic fibroblast growth factor (bFGF) can inhibit endoplasmic reticulum stress induced by IS through activating PI3K/Akt and ERK1/2 pathways [48]. In addition, it was also found that that Aloperine (ALO) can improve cerebral I/R injury and alleviate oxidative stress via activating the PI3K/AKT signaling pathway in order to support the therapeutic potential of ALO against cerebral I/R injury in IS [49]. Similarly, the NX210 peptide can prevent oxidative stress and neuronal apoptosis in cerebral I/R via up-regulation of the Integrin-beta1/PI3K/Akt signaling pathway [50]. Moreover, total flavonoids of Chuju (TFCJ) may also reduce oxidative stress and apoptosis of IS by activating the PI3K/Akt/mTOR signal pathway [51]. In conclusion, the above evidence suggests that drugs can ameliorate oxidative stress after IS through the PI3K/Akt signaling pathway.

#### 2.2.5. Autophagy Caused by Ischemic Stroke with PI3K/Akt Signaling Pathway 

Autophagy is a catabolic pathway for cell self-protection, which achieves self renewal by decomposing damaged nerve cells [52]. In response to stress changes in the internal and external environment of the body, autophagy rapidly enters a bidirectional regulation state after IS [53]. Firstly, moderate autophagy can transform its degradation products and synthesize adenosine triphosphate (ATP) to provide energy for nerve cells. Secondly, excessive autophagy caused by increased ischemia will lead to the autophagic death of nerve cells [54]. 

Plenty of studies have shown that the PI3K/Akt signaling pathway is closely related to autophagy. One study showed that diosgenin injected intraperitoneally with 100mg/kg can improve LC3-II/LC3-I, Beclin-1 and IL-1 of I/R model rats by activating the PI3K/Akt signal pathway in order to reduce autophagy and play a neuroprotective role [55]. Additionally, mTOR is the downstream target of PI3K/Akt signaling pathway, can regulate a variety of intracellular signals and have an effect in autophagy [56]. A nested case–control study showed that the expression of mTOR protein in IS was significantly increased and the activation of the PI3K/Akt/mTOR signaling pathway autophagy signaling pathway was enhanced, which inhibited excessive autophagy of nerve cells [57]. It has been reported that the TCM prescription Tong-Qiao-Huo-Xue decoction can activate PI3K/Akt/mTOR signaling pathway to reduce brain microvascular endothelial cell (BMECs) autophagy after cerebral ischemia/reperfusion injury [58]. Similarly, Ibrutinib also can ameliorate autophagy and cerebral ischemia/reperfusion injury through the PI3K/Akt/mTOR signal pathway [59]. Moreover, inhibition of miR-124 can increase the PI3K/Akt/mTOR signaling pathway, thus inhibiting cell apoptosis and autophagy caused by IS [60]. Interestingly, we found that EA can also inhibit neuronal autophagy and apoptosis via the PI3K/Akt signaling pathway following IS, and it is a safe and effective therapy for ischemic stroke in both clinical and laboratory settings [41]. Therefore, the above evidence indicates that the improvement of IS may be achieved by activating the PI3K/Akt signaling pathway to reduce excessive autophagy. 

Overall, IS is formed by the joint action of many pathological factors. Drugs activate the PI3K/Akt signaling pathway in the early stage of IS development, then crosstalk cell apoptosis, autophagy, oxidative stress, ischemia-induced inflammation, and other important mechanisms. These mechanisms trigger a series of cascade reactions, which initiate the protective effect on damaged nerve cells. However, it should be noted that these effects after IS do not exist alone, but overlap. In order to make better use of the PI3K/Akt signaling pathway, multi-target and multifaceted comprehensive interventions should be adopted in the treatment of IS.

## 3. Natural Medicine for the Treatment of Ischemic Stroke through PI3K/Akt Signaling Pathway

### 3.1. Herbal Medicine

After long-term research, it has been found that the following representative herbal medicine has significant therapeutic effects on ischemic stroke and the accumulated data confirm that the effects of these medicines are likely to be mediated by PI3K/Akt signaling pathway. In this section, we reviewed the effects of herbs and their active ingredients on IS and their potential mechanism on the PI3K/Akt signaling pathway (Table 1).

#### 3.1.1. *Chuanxiong*

*Chuanxiong* (CX, Family: Umbelliferae) is the dried rhizome of *Ligusticum chuanxiong* Hort. CX has the effect of promoting blood circulation. It contains volatile oil, alkaloids, and other active ingredients. Among them, the most studied is the active component of alkaloid: ligustrazine [61]. It has been proven that ligustrazine can reduce the infarct area and water content of cerebral ischemia–reperfusion rats and improve the neurological function. It also can down-regulate the expression of Bax mRNA in hippocampal neurons and inhibit neuronal apoptosis [62]. Additionally, ligustrazine can control blood flow velocity, promote blood circulation, and play the role of anti-platelet aggregation and anti-thrombosis [63]. Additionally, the vitro experiments have proved that ligustrazine may have a significant neuroprotective effect on PC12 cells induced by glucose deprivation injury with inhibiting Bax/Bcl-2 and Caspase-3 apoptosis pathways [62]. It can be seen that ligustrazine may have a good effect on improving IS.

Various studies have reported that ligustrazine is related to the PI3K/Akt signaling pathway. It has been confirmed that ligustrazine may inhibit the proliferation and inflammation of rat pulmonary vascular smooth muscle cells (PASMCs) by activating PI3K/Akt signaling pathway, thus alleviating monocrotaline (MCT)-induced pulmonary hypertension in rats [64]. Additionally, ligustrazine can activate autophagy by regulating the PI3K/Akt/mTOR signaling pathway, so as to improve the neurocognitive impairment induced by endotoxin and reduce neuronal damage [65]. Ligustrazine can also play a neuroprotective effect on brain I/R injury in rats by activating the PI3K/Akt signaling pathway [66]. This evidence suggests that ligustrazine may improve IS-induced neuronal inflammation, autophagy, and apoptosis by regulating the PI3K/Akt signaling pathway. 

Moreover, ligustrazine is also widely used in the clinic. Compared to conventional medicine treatment alone, ligustrazine plus conventional medicine treatment showed a significant difference in the reduction of stroke recurrence either at the end of 1-year follow-up or 3-years observation. The ligustrazine group showed a higher survival rate and significantly better effective rate than that of the control group at the end of a 1-year visit [67]. Thus, ligustrazine is a drug with high safety. Furthermore, ligustrazine comes from CX, which means that CX may be a potential drug for the treatment of IS. It can be confirmed by further study.

#### 3.1.2. *Salvia miltiorrhiza*

*Salvia miltiorrhiza* (SM, Family: Labiatae) is the dried roots and rhizomes of *Salvia miltiorrhiza* Bunge. SM has the effect of promoting blood circulation, dispersing blood stasis, and relieving pain. The active ingredients include tanshinone, danshensu, and salvianolic acid. Studies have shown that SM possess good anti-inflammatory, antioxidant, and anti-apoptosis activities [68]. A large number of data show that there are three salvianolic acids have good therapeutic effect on IS. Salvianolic acid A (Sal A) can inhibit inflammation and apoptosis in mice through PI3K/Akt signaling pathway, and reduce nerve injury caused by IS [69]. Salvianolic acid B (Sal B) can induce the phosphorylation of Akt and mTOR and promote cell migration and increase angiogenesis in IS rats, thus reducing neuronal apoptosis and increasing the expression of stanniocalcin-1 (STC1) [70]. Additionally, Sal B also can maintain neural stem/progenitor cells self-renewal and promote proliferation through the PI3K/Akt signaling pathway, and improve cognitive impairment after IS in rats [71]. Based on the mechanism of network pharmacology and molecular docking, it has been revealed that salvianolic acid C (Sal C) may have a protective effect on the brain of IS rats through the PI3K/Akt signaling pathway [72]. Moreover, tanshinone is also associated with the PI3K/Akt signaling pathway and IS. It has been reported that tanshinone IIA (Tan IIA) may protect the neurons of primary cultured cortical neurons through PI3K/Akt signaling pathway [73]. Similarly, sodium Tan IIA sulfonate (STS) can prevent progressive and persistent brain injury and improve neurological dysfunction by activating the PI3K/Akt signaling pathway in rats with middle cerebral artery occlusion/reperfusion (MCAO/R) [74]. Additionally, in clinical treatment, the use of STS after stroke can improve the neurological function of patients with acute IS after recombinant tissue plasminogen activator (rt-PA) treatment by reducing BBB leakage and damage [75]. This evidence suggests that *Salvia miltiorrhiza* is not only safe in the treatment of IS, but also has a variety of compounds that can be studied. It is a potential drug for the treatment of IS.

#### 3.1.3. *Radix Angelicae sinensis*

*Radix Angelicae sinensis* (RAs, Family: Umbelliferae) is the root of *Angelica sinensis* (Oliv.) Diels. RAs has a great effect of tonifying blood, regulating menstruation, and relieving pain. The main active ingredients are volatile oil, polysaccharides, and organic acids. Among them, angelica polysaccharides (ASP) has been proved to have the effect of anti-cerebral ischemia. It can increase the antioxidant activity of cortical neurons and the number of microvessels, thus improve the blood flow after cerebral ischemia [76]. Additionally, it has been reported that ASP can effectively improve neural function and neuronal cell apoptosis in CIRI rats by activating PI3K/Akt signaling pathway [77].

In addition, ferulic acid and ligustilide, the effective ingredients of angelica volatile oil have the effect of ameliorating IS. They have significant effects of anti-inflammation, antioxidation, angiogenesis, nerve regeneration, anti-platelet aggregation, anti-atherosclerosis, and protection of blood vessels. It is helpful to improve the neural function of IS [78]. According to prediction and docking simulation, it is shown that ferulic acid can synergistically affect IS through genes such as met-enkephalin (MEK) and NF-κB in PI3K, Akt cascade signals [79]. As for ligustilide, it has been found to attenuate hippocampal neuronal apoptosis induced by ischemia–reperfusion by activating PI3K/Akt signaling pathway [80]. The above evidence shows that the mechanism of *Radix Angelicae sinensis* in treating IS is by activating PI3K/Akt signal pathway.

#### 3.1.4. *Astragalus membranaceus*

*Astragalus membranaceus* (AM, Family: Leguminosae) is the root of *Astragalus memeranaceus* (Fisch.) Bge. Var. *mongholicus* (Bge.) Hsiao. It has the functions of promoting diuresis, detumescence, blocking arthralgia, supporting toxin, and expelling pus. Additionally, the main active ingredients of AM are astragalus polysaccharides, saponins, and flavonoids. Studies have shown that AM has a wide range of effects on blood vessels. It includes reducing the inflammatory response after cerebral ischemia–reperfusion, scavenging oxygen free radicals, protecting vascular endothelial cells, reducing vascular permeability, increasing cerebral blood flow, and inhibiting apoptosis [81]. 

Astragaloside IV (AS-IV) is the main saponins of *Astragalus membranaceus*. AS-IV has been reported to promote the cell viability of HT22 after hypoxic glucose deprivation/reperfusion (OGD/R). It can also balance the expression of Bcl-2 and Bax in vitro and increase the expression of LC3II/LC3I. This indicate that AS-IV plays a neuroprotective role by down-regulating apoptosis by promoting the degree of autophagy and improve IS [82]. Moreover, in vivo and in vitro experiments have proved that AS-IV can down-regulate the expression of IL-17 protein, regulate cell apoptosis by regulating PI3K/Akt/GSK-3β signaling pathway, and reverse the promoting effect of IL-17 on the proliferation, neural regeneration, and cognitive dysfunction of neural stem cells (NSCs) after IS [83]. In addition, it was found that after administration of total flavonoids of AM in rats with cerebral ischemia, the expression of MDA, IL-1 β, TNF-α, Caspase-3, and Bax mRNA decreased significantly, while the activity of Bcl-2 mRNA, PI3K, and Akt protein increased significantly. It is suggested that the protective effect of total flavonoids of AM on cerebral ischemia–reperfusion injury is related to the anti-apoptotic effect caused by activating PI3K/Akt signaling pathway [84]. 

Moreover, Astragalus polysaccharides (APS) is another important active ingredient in *Astragalus membranaceus*. Studies have shown that APS exerts anti-Parkinson via activating the PI3K/Akt/mTOR signaling pathway to increase cellular autophagy levels in vitro [85]. APS can also reduce the content of TNF-α by activating the PI3K/Akt signaling pathway [86]. This suggests that APS may improve autophagy and apoptosis of IS through PI3K/Akt signaling pathway.

In conclusion, AM has great development potential as a commonly used natural medicine. The study of AM to improve IS through the PI3K/Akt signaling pathway provides a new idea for the treatment of IS.

#### 3.1.5. *Safflower*

*Safflower* (SF, Family: Compositae) is the flower of *Carthamus tinctorius* L. The main active ingredients of SF are alkaloids, flavonoids, and organic acids. These ingredients can effectively prevent and treat cardiovascular and cerebrovascular diseases such as atherosclerosis and thrombosis. 

Safflower yellow (SY) is one of the effective flavonoids of SF, which is widely used for the treatment of acute ischemic stroke in China [87]. Studies have shown that SY can significantly reduce cerebral infarction area, increase cerebral vascular density, and reduce permeability. Finally, SF treatment leads to effective improvement of brain edema [87] and SY can also inhibit the formation of thrombus by inhibiting platelet activation and aggregation. Reduce brain injury caused by reperfusion by reducing blood viscosity, dilating blood vessels, and improving blood circulation [88]. Therefore, SY is widely concerned in the treatment of IS and emerging evidence has shown that SY may inhibit the recovery of cell proliferation by TNF-α through PI3K/Akt signaling pathway [89].

Hydroxysafflor yellow A (HSYA), a major active ingredient of the SF, has drawn more interest in recent year for its multiple pharmacological actions in the treatment of cerebrovascular and cardiovascular diseases [90]. It is reported that HSYA may inhibit foam cell formation and vascular endothelial cell dysfunction and inhibit the proliferation and migration of vascular smooth muscle cells through PI3K/Akt/mTOR signaling pathway. It can also inhibit platelet activation, thus reducing a series of injuries caused by IS [91]. Moreover, HSYA may play a protective role and inhibit apoptosis in focal cerebral ischemia model rats by activating PDGF-mediated PI3K/Akt signaling pathway [92]. Additionally, in clinical treatment, HSYA was safe and well-tolerated at all doses for treating IS patients with blood stasis syndrome (BSS). The medium (50 mg/d) or high dose (75 mg/d) might be the optimal dose for a phase III trial [93].

#### 3.1.6. *Ginkgo biloba leaf*

*Ginkgo biloba leaf* (GBL, Family: Ginkgoaceae) are the dried leaves of *Ginkgo biloba* L. GBL is one of the oldest species in the world, known as “A living fossil full of treasure”. Ginkgo flavonoids and ginkgolides are the main active ingredients of GBL, which have certain therapeutic and preventive effects on ischemic cerebrovascular diseases. 

Studies have shown that ginkgolides can reduce necrotic neurons and hippocampal lesions, reduce blood viscosity, dilate blood vessels, and increase cerebral blood flow [94]. Such as ginkgolide A (GA) has been shown to inhibit lipopolysaccharide (LPS)-induced inflammation in human coronary artery endothelial cells (HCAECs), and its anti-inflammatory activity may be related to the inhibition of PI3K/Akt signaling pathway [95]. Additionally, mechanism studies confirmed that ginkgolide B (GB) methane-sulfonate has angiogenic effect in vivo and in vitro through PI3K/Akt/ glycogen synthase kinase 3 (GSK3) signaling pathway. It is a potential anti-chronic IS drug [96]. Moreover, the combination of ginkgo flavonoids and ginkgolides can significantly reduce brain edema and antagonize platelet activating factor in order to reduce the volume of cerebral infarction and nerve injury [97]. Additionally, the combined application of the two can improve cerebral I/R injury through the PI3K/Akt/Nrf2 signaling pathway and multi-component intrabody process [98]. Furthermore, the existing clinical evidence shows that GBL preparations (GLP) have a good therapeutic effect on patients with IS and can improve their hemorheology indices. Moreover, GLP is shown to be relatively safe [99]. 

#### 3.1.7. *Erigeron breviscapus*

*Erigeron breviscapus* (EB, Family: Compositae) is the whole grass of *Erigeron breviscapus* (Vant.) Hand.-Mazz. EB contains flavonoids, caffeic acid esters, aromatic acids, coumarins, pyranones, and other ingredients. Among them, flavonoids scutellarin (also known as breviscapine) and caffeic acid esters are the main active ingredients. 

Scutellarin, the main bioactive flavonoid glycoside extracted form EB, has been reported to exert positive effects on anti-inflammatory reactions. It has been reported that scutellarin can regulate osteoarthritis through PI3K/Akt/mTOR signaling pathway in vitro [100]. Additionally, scutellarin can also inhibit epithelial-interstitial transformation and angiogenesis through PI3K/Akt/mTOR signaling pathway [101]. Meanwhile, scutellarin may play a therapeutic role in IS through PI3K/Akt/mTOR signaling pathway [102]. This evidence shows that scutellarin can improve the inflammatory response and neuronal apoptosis of IS through PI3K/Akt signaling pathway.

Emerging evidence shows that EB injection and scutellarin can inhibit the activation of MMP-9 and reduce the degradation of claudin-5 by inhibiting the synthesis of inducible nitric oxide synthase (iNOS). Finally, the structural integrity of BBB after ischemia was shown to be protected [103]. Moreover, the network pharmacological studies confirmed that EB may improve brain barrier damage caused by IS through PI3K/Akt signaling pathway [103]. Thus, it can be seen that EB has great potential in the treatment of IS. However, there is a lack of basic research on EB through PI3K/Akt signaling pathway in the treatment of IS. It is suggested that EB may become a new research breakthrough point.

#### 3.1.8. *Ginseng*

*Ginseng* (Gs, Family: Araliaceae) is the root of *Panax ginseng* C.A. Meyer. The main active ingredient of *Ginseng* is ginsenoside (GS), which have always been referred to as “all-healing” and widely used for its extensively medicinal value. GS have diverse biological activity which might be related to inflammation, apoptosis, oxidative stress, and angiogenesis [104]. 

There are several types of GS that have been studied. It has been reported that ginsenoside-Rb1 (GS-Rb1) can protect the integrity of blood–brain barrier in IS by inhibiting free radicals derived from matrix metalloproteinase-9 and NADPH oxidase 4 (NOX4) induced by neuroinflammation [105]. GS-Rb1 can also inhibit autophagy of cardiomyocytes and reduce myocardial I/R injury, and the mechanism is mediated by the activation of the PI3K/Akt/mTOR signaling pathway [106]. In addition, ginsenoside-Rg1 (GS-Rg1) has been shown to promote cerebral angiogenesis through the PI3K/Akt/mTOR signaling pathway in recent years [107]. GS-Rg1 can also prevent cognitive impairment induced by isoflurane anesthesia in aged rats through antioxidant, anti-inflammatory, and anti-apoptotic effects mediated by the PI3K/Akt/GSK-3 signaling pathway [108]. Moreover, GS-Rd can reduce Tau phosphorylation after transient forebrain ischemia through the PI3K/Akt/GSK-3β signaling pathway. Moreover, the clinical studies showed that GS-Rd improves the main prognosis of acute IS and has an acceptable profile of adverse events [109]. Emerging evidence proves that GS-Rh1 exerts neuroprotective effects by activating the PI3K/Akt pathway in amyloid-beta induced SH-SY5Y cells [110]. GS can not only alleviate cerebral I/R injury in rats, but also activate PI3K/Akt and extracellular signal-regulated kinase 1/2 (ERK1/2) pathways and promote neurogenesis by increasing the expression of VEGF and brain-derived neurotrophic factor (BDNF) [111]. In conclusion, Gs can reduce the inflammatory reaction and apoptosis caused by IS in multi-angle by activating the PI3K/Akt signaling pathway. It has various active ingredients and is a high-quality natural medicine worthy of further study.

#### 3.1.9. *Radix Paeoniae Rubra*

*Radix Paeoniae Rubra* (RPR, Family: Ranunculaceae) is the dried root of *Paeonia lactifloral* Pall, or *Paeonia veitchii* Lynch. The main active ingredient of RPR is glycosides, which have the effects of clearing heat and cooling blood, promoting blood circulation, and dispelling blood stasis. The role of paeoniflorin (PF) is particularly prominent. Studies have shown that PF is an active monomer that can promote vascular endothelial progenitor cells and angiogenesis in IS rat model [112]. PF is also the main anti-inflammatory component in RPR, which can inhibit the production of pro-inflammatory mediators such as TNF-α and IL-1β. It has been reported that PF improves functional recovery through repressing neuroinflammation and facilitating neurogenesis in rat IS model [113]. Additionally, the combination of PF and calycosin-7-glucoside can alleviate IS injury via the PI3K/Akt signaling pathway [114]. Clinical studies showed that the levels of apoptosis and inflammatory factors IL-1β decreased in PF treatment group. It is suggested that PF is a promising method for the treatment of brain I/R injury. However, additional preclinical studies are needed to more accurately evaluate the efficacy and safety of PF [115]. 

#### 3.1.10. *Panax notoginseng*

*Panax notoginseng* (PN, Family: Araliaceae) is the dried root of *Panax notoginseng* (Burk.) F.H. PN has a variety of physiologically active ingredients, such as total saponins, volatile oil, amino acids, flavonoids, and polysaccharide. 

*Panax notoginseng* saponins (PNS) is one of the ingredients with rich pharmacological effects. It can promote the mechanism of neural function recovery, and plays an important role in brain protection after cerebral infarction [116]. Additionally, it can also reduce leukocyte-mediated microvascular disturbance during the onset of IS [117]. According to the observed results, PNS may be an exogenous regulatory factor that activates the Nrf2 antioxidant signal through the PI3K/Akt signaling pathway to protect against OGD/R-induced BBB damage in vitro [118]. Moreover, PNS can also protect cardiomyocytes from apoptosis induced by ischemia in vivo and in vitro by activating PI3K/Akt signaling pathway [119]. Additionally, other studies have shown that PNS Rb1 can improve cognitive and sensorimotor disorders at least in part by regulating the Akt/mTOR/PTEN signaling pathway [120]. 

In clinical treatment, patients who received PNS combined with conventional treatments (CTs) showed significantly high improvements in neurological function among individuals with IS on the neurologic deficit score. Compared with CTs alone, PNS can significantly improve the overall response rate (ORR). In addition, the incidence of adverse reactions in PNS group was lower than that in control group and all of them were mild adverse. 

In conclusion, the above evidence means that PNS may be effective and safe in treating IS on ameliorating the neurological deficit, improving activities of daily living function, and enhancing antiplatelet effects, and its mechanism is likely to be mediated by PI3K/Akt signaling pathway. However, more high-quality evidence is needed before it can be recommended for routine antiplatelet therapy in patients with IS [121]. 

### 3.2. Herbal Prescriptions 

All the above natural medicines have special application in China, namely traditional Chinese medicine (TCM) prescription. TCM prescription combines effective natural medicines to treat related diseases, and each medicine can interact with each other to achieve the best therapeutic effect. TCM prescription has the characteristics of multi-target, multi-component, and multi-mechanism, which plays a synergistic effect through multiple mechanisms to produce a better effect in the treatment of cerebral ischemia. In this section, we reviewed the representative TCM prescriptions, thus revealing the relationship between it and the PI3K/Akt signaling pathway. 

#### 3.2.1. Buyang Huanwu Decoction 

Buyang Huanwu Decoction (BHD) is composed of *Astragalus membranaceus*, *Angelica sinensis*, *Radix Paeoniae Rubra*, *Chuanxiong*, *Safflower,* and *Peach kernel*. BHD is commonly used in the clinical treatment of IS, coronary heart disease, and other vascular embolic diseases. The active ingredients in BHD play a potential pharmacological role in the treatment of IS by regulating the following multiple targets: (1) Propyl gallate, calycosin-7-O-beta-D-glucoside, paeonol, and ferulic acid can significantly inhibit apoptosis; (2) Propyl gallate and formononetin can significantly inhibit LPS-induced NO release; (3) HSYA and inosine can protect cells against the injuries caused by glutamate; and (4) Formononetin, AS-IV, astrISoflavan-7-O-beta-Dglucoside, inosine, PF, ononin, paeonol, propyl gallate, ligustrazine, and ferulic acid can significantly suppress the constriction of the thoracic aorta induced by KCl in rats [122]. In addition, the previous study confirmed that the ingredients of BHD may antagonize HT22 cell injury induced by cerebral I/R injury and oxygen-glucose deprivation in rats in vitro by reducing the inflammatory reaction and apoptosis, which is a novel IS protection strategy [123]. 

BHD is also closely related to the PI3K/Akt signaling pathway. It can promote vascular endothelial growth factor receptor-2 (VEGFR2) phosphorylation through the PI3K/Akt signaling pathway to restore neurological function and angiogenesis in mice with intracerebral hemorrhage, thus alleviating stroke [124]. Further studies have shown that BHD can significantly up-regulate p-PI3K, p-Akt, and phosphorylated Bcl-2-related death promoter (p-Bad), both in vivo and in vitro. It is suggested that BHD can achieve neuroprotection and promote nerve regeneration by activating PI3K/Akt/Bad signaling pathway, finally ameliorate IS [125]. 

In addition, BHD is a TCM prescription widely used in the rehabilitation of patients with IS in China. However, the methodological quality of the research it incorporates is relatively low [126]. The clinical efficacy and safety of BHD need to be well designed to ensure the clinical recommendation of BHD for the rehabilitation of patients with ischemic stroke.

#### 3.2.2. Taohong Siwu Decoction 

Taohong Siwu Decoction (THSWD) comes from “The Golden Mirror of Medicine”. It is composed of *Peach kernel*, *Safflower*, *Radix rehmanniae*, *Radix Angelicae sinensis*, *Peony,* and *Chuangxiong*. The main active ingredients of THSWD are HSYA, PF, paeonol, ligustrazine, and ferulic acid [127]. 

Studies have shown that THSWD can reduce the release of inflammatory factors and inhibit the activation of the complement signaling pathway, thus preventing IS [128]. Additionally, THSWD can also reduce the expression of the TNF-α pathway and reduce the necrosis of brain cells after cerebral ischemia, thus protecting the brain tissue of rats [129]. Moreover, we found that THSWD has a significant effect on the rat model of cerebral I/R injury. The mechanism may be to activate the PI3K/Akt signaling pathway to promote neovascularization after cerebral ischemia and restore neurological function in rats [130]. These results suggest that THSWD can reduce IS damage through the PI3K/Akt signaling pathway and may have a preventive effect. However, there is no clinical study on the treatment of IS with THSWD, which needs to be further confirmed. 

#### 3.2.3. Xiaoyao San

Xiaoyao San (XYS) comes from “Prescription of peaceful benevolent dispensary”. It is composed of *Licorice*, *Radix Angelicae sinensis*, *Poria*, *Largehead atractylodes*, *White peony*, and *Radix bupleuri*. The active ingredients of XYS contain saikoside, ferulic acid, ligustilide, atractylenolide, PF, albiflorin, liquiritin, glycyrrhizic acid, and pachymic acid [131]. XYS has the effect of soothing the liver and relieving depression, nourishing blood, and invigorating the spleen.

Emerging evidence demonstrates that XYS exerts anti-apoptotic and neuroprotective effects by activating the PI3K/Akt signaling pathway to inhibit Caspase-3 and Caspase-9 mediated apoptosis cascades, thus achieving the therapeutic effect of IS. XYS also has a good clinical effect in treating IS, with few adverse reactions [132].

#### 3.2.4. Danhong Injection

Danhong injection (DHI) is composed of *Danhong* and *Safflower*. DHI has the effect of promoting blood circulation and removing blood stasis, dredging pulse, and relaxing collaterals. The active ingredients of DHI are ferulic acid, cryptotanshinone, quercetin, and anhydrosafflor yellow B [133]. 

Studies have shown that DHI can reduce cerebral edema and preserve ischemic penumbra after cerebral ischemia–reperfusion [134]. Additionally, DHI can also significantly down-regulate the phosphorylation levels of NF-κB and MAPK pathway proteins in ischemic brain tissue. It is suggested that DHI has an anti-neuroinflammatory effect and is helpful to improve the injury of central nervous system after IS [135]. It is worth noting that DHI is closely related to the PI3K/Akt signaling pathway. Studies have shown that DHI can significantly reduce the expression of pro-apoptotic factors Bad, Bax, and Bcl-2 through the PI3K/Akt signaling pathway, while the up-regulate the expression of anti-apoptotic factor Bcl-2 [136]. This evidence suggests that DHI may reduce the inflammatory response and apoptosis induced by IS through PI3K/Akt signaling pathway.

In addition, there are many clinical studies on the treatment of IS with DHI. Studies have shown that DHI injection combined with pivastatin can effectively inhibit inflammatory reaction and oxidative stress injury in patients with IS, and promote the recovery of neurological function [137]. Moreover, DHI has an important clinical improvement role in the NIHSS score, and no adverse events have been observed [138]. This evidence indicates that DHI is effective and safe in the treatment of IS.

#### 3.2.5. Sanhua Decoction

Sanhua decoction (SHD) comes from “The Yellow Emperor’s Classic of Medicine”. It is composed of *Rhubarb*, *Fructus aurantii immaturus*, *Magnolia officinalis*, and *Notopterygium.* SHD mainly contains flavonoids, anthraquinones, coumarins, phenylpropanoid glycosides, alkaloids, and lignans [139]. 

Research shows that SHD exerts neuroprotection through regulating the phosphorylated Tau level and promoting adult endogenous neurogenesis after cerebral I/R injury [140]. Additionally, SDH can also increase the levels of p-PI3K and p-Akt, and reduce the expression of tumor necrosis factor-α and IL-6 [141]. It is suggested that SDH may have a protective effect on brain injury caused by IS by regulating the PI3K/Akt signaling pathway and tumor necrosis factor. These conclusions provide a theoretical basis for the development of SHD as a new drug for the treatment of IS. 

Clinically, SHD is also widely used to treat IS, especially in reducing blood viscosity and regulating circulatory disturbances, with almost no adverse reactions [141].

#### 3.2.6. Xingnaojing Injection

Xingnaojing injection (XNJI) is derived from Angong Niuhuang Wan. It is composed of *Turmeric*, Moschus, Borneolum syntheticum, and *Gardenia jasminoides*. The main active ingredients of XNJI are muscone, camphor, curcumin A, curcumin B, curcumone, and curcumdione [142]. XNJI has the effect of cooling blood, activating blood circulation, opening orifices, and awakening the brain. 

XNJI has been shown to prevent apoptosis of human microvascular endothelial cells through the PI3K/Akt/endothelial nitric oxide synthase (ENOS) pathway in a rat model of IS [143]. In addition, muscone is one of the active ingredients in XNJI. It has also been shown to significantly reduce neuronal apoptosis through the PI3K/Akt signaling pathway [144]. It indicates that XNJI may reduce oxidative stress and neuronal apoptosis induced by IS through PI3K/Akt signaling pathway.

Clinically, the reports showed that XNJI appears to be effective and safe for the emergency treatment of IS. The first 6 h after IS may be the best time to start taking medication [145]. However, more high quality randomized controlled trials are needed to determine the appropriate start-up time. 

### 3.3. Animal Medicine

Most of the animal medicine used for the treatment of IS is insect medicine. Among them, earthworm, leech, and scorpion can inhibit thrombosis, lipid lowering, and platelet aggregation in varying degrees. Additionally, they can also improve the cerebral blood circulation and increase the blood oxygen supply in the cerebral ischemic area. Finally, they can reduce brain injury and promote the recovery of nerve cell function [146]. In this section, we reviewed the representative insect medicine and prescriptions related to PI3K/Akt signaling pathway in the therapeutics of IS (Table 2).

#### 3.3.1. *Earthworm*

*Earthworm* has the effects of clearing heat, calming shock, dredging collaterals, relieving asthma, and diuresis. It is mostly used for high fever, epileptic convulsions, limb paralysis, and hemiplegia. Modern studies have found that earthworms also have antithrombotic, immune, and anti-inflammatory effects. In addition, it is an important animal medicine in China’s current guidelines for the treatment of IS.

The main active ingredient of earthworm is lumbrokinase (LKs). It is an oral supplement to support and maintain healthy cardiovascular function and has a history of clinical treatment of cardiovascular disease for more than 10 years. Studies have shown that LKs treatment after ischemia can reduce myocardial I/R injury by activating SIRT1 signal pathway, thus reducing oxidative damage, inflammatory reaction and apoptosis [147]. Additionally, LKs can decrease the protein levels of caspase-8, caspase-9, and caspase-3 by enhancing the expression of survival-related proteins PI3K/Akt and Bcl-2 [148]. In addition, emerging evidence shows that earthworm extract can improve neurological dysfunction, histomorphology, and neuronal damage in mice with cerebral ischemic penumbra. It also can up-regulate the expression of PI3K and Bcl2 protein and increase the ratio of p-Akt/Akt [149]. It is suggested that earthworm may improve the neuronal damage in the penumbra of cerebral ischemia in IS mice, and its mechanism may be through the activation of the PI3K/Akt signaling pathway.

The report confirmed that LKs can also degrade fibrin and enhance the function of vascular endothelial cells during the clinical treatment of IS, thus significantly improving the hemorheology of patients with IS [150]. Moreover, LKs combined with aspirin has been shown to be more effective in reducing recurrence of IS than aspirin alone without increasing the incidence of massive bleeding and with higher safety [151]. The above results provide direct data for Chinese clinicians to choose the treatment plan of IS. 

#### 3.3.2. *Leech*

*Leech*, which has been recorded in the “Sheng Nong’s herbal classic”. It has high medicinal value and generally grows and breeds in inland fresh water. The main active ingredient of leech can be divided into two categories: one directly acts on the coagulation system such as hirudin, and the other is protease inhibitors such as fibrinolysin. Among them, hirudin has been widely used in stroke. Natural hirudin is the strongest specific thrombin inhibitor found so far. It has antithrombotic effect with low risk of bleeding, so it is widely used in clinic [152]. 

Hirudin protects BV-2 microglia against OGD/R and inhibits neuroinflammation mediated by Nod-like receptor protein 3 (NLRP3) inflammatory bodies. It is a promising early intervention drug for acute ischemic stroke [153]. Studies have shown that hirudin can reduce inflammatory cell infiltration and interstitial collagen accumulation through the PI3K/Akt signaling pathway [154]. Additionally, it has been reported that hirudin may protect the kidney by ameliorating renal autophagy impairment through modulating the PI3K/Akt pathway [155]. These results suggest that hirudin may attenuate IS-induced inflammation and autophagy through the PI3K/Akt signaling pathway.

Moreover, clinical treatment results of a study showed that hirudin combined with aspirin significantly reduced the safety risk of secondary prevention of non-valvular atrial fibrillation thrombotic stroke, indicating that hirudin has great potential in clinical treatment of IS [156].

#### 3.3.3. *Scorpion*

At present, the global pharmacological research on scorpion is mainly focused on anti-thrombosis, anticoagulation, fibrinolysis, analgesia, anti-epilepsy, anti-tumor, and immune regulation. Low molecular compounds with anticoagulant activity have been isolated from scorpion venom, and their structures have been determined to be adenosine, dipeptide Leu Trp and Ile Trp [157]. 

Adenosine is a platelet agglutination inhibitor with anticoagulant properties [158]. Studies have shown that adenosine system plays a positive role in the treatment of IS and may open the way for the development of more targeted treatments and biomarkers for IS [159]. In addition, it has been confirmed that scorpion venom polypeptide extract (PESV) may inhibit the proliferation of K562 cells by regulating PI3K and p-Akt signal proteins, which has been involved in the development of prostate cancer [160]. Combined with the remarkable therapeutic effect of scorpion on stroke, its mechanism of the PI3K/Akt signaling pathway could be further studied.

### 3.4. Commonly Used Prescriptions of Traditional Chinese Medicine Containing Insects

#### 3.4.1. Tongxinluo Capsule

Tongxinluo capsule (TXL) is composed of *Ginseng*, *Leech*, *Scorpion*, *Ground Beetle*, *Centipede*, *Cicadae Periostracum*, *Radix Paeoniae Rubra*, and Borneol. The main active ingredients are GS, hirudin, and PF. TXL has been widely used in China for the treatment of acute stroke and for neuroprotection. 

Studies have shown that TXL can protect the blood-brain barrier after stroke by inhibiting lipoprotein receptor-related protein 1 (LRP-1) pathway [161]. After TXL treatment, the loss of neurons in prefrontal cortex was significantly reduced and the neurological function was significantly improved [162]. Additionally, TXL also alleviates cerebral microcirculatory disturbances against ischemic injury by modulating endothelial function and inhibiting leukocyte-endothelial cell interactions [163]. We also found that TXL can significantly up-regulate the expression of phosphoinositide-dependent kinase 1 (p-PDK1) and p-Akt by activating PI3K/Akt signaling pathway, which has a neuroprotective effect on brain I/R injury and neuronal apoptosis [164]. This suggests that TXL may improve IS-induced apoptosis and brain damage through PI3K/Akt signaling pathway.

TXL is superior to conventional treatment in improving clinical overall response rate and hemorheological indexes and is relatively safe. Due to the lack of existing research, more well-designed high-quality research is needed for further verification [165].

#### 3.4.2. Naoxintong Capsule

Naoxintong capsule (NXT) is a clinical prescription for ischemic cerebrovascular disease, which is mainly composed of *Scorpion*, *Leech*, *Salvia miltiorrhiza*, *Achyranthes bidentata*, *Frankincense*, *Myrrh*, *Suberect Spatholobus Stem*, and *Mulberry branch*. The main active ingredients are adenosine, hirudin, tanshinone, danshensu, and salvianolic acid. 

Emerging evidence has shown that NXT can effectively inhibit inflammation and oxidative stress, and reduce other pathological injuries caused by cerebral ischemia cascade reaction. Additionally, NXT can inhibit the expression of NF-κB and down-regulate the expression of inflammatory cytokines IL-1β, IL-6, and TNF-α. Thus, PI3K/Akt signaling pathway is activated, which plays anti-inflammatory, antioxidant, and cell protective roles [166,167]. This indicates that NXT may ameliorate IS-induced cellular inflammation, oxidative stress, and brain damage through the PI3K/Akt signaling pathway.

In clinical use, it has been found that NXT combined with aspirin may provide an alternative drug for the prevention of IS in elderly nonvalvular atrial fibrillation patients who are not tolerant to warfarin [168]. Additionally, a meta-analysis also confirmed the reliability of NXT in the efficacy and safety of IS [169].

#### 3.4.3. Shuxuetong Injection 

Shuxuetong (SXT) injection is a preparation of animal traditional Chinese medicine composed of hirudin in leech and lumbrokinase in earthworm. SXT has the effect of promoting blood circulation and removing blood stasis, and has been widely used in the treatment of IS. 

An experimental study in vitro confirmed that SXT can inhibit the production of reactive oxygen species and mitochondrial superoxide by increasing the activity and expression of Bcl-2. It can also activate NF-κB and inhibit the expression of pro-inflammatory factors [170]. In addition, SXT can also restore and adjust the metabolic disorder of cerebral cortex and hippocampus in brain I/R rats [171]. Moreover, SXT has been proven to have the effect of promoting angiogenesis and improving cerebral perfusion [172]. Moreover, the exact target pathway of SXT in the treatment of IS has been confirmed by experimental studies. SXT can repair cerebral ischemic injury in rats by using the PI3K/Akt signaling pathway, MAPK signal pathway, HIF-1 signal pathway, and metabolic pathway-related proteins as targets [172]. Combined with the above evidence, it is suggested that SXT may alleviate IS-induced apoptosis and oxidative stress through the PI3K/Akt signaling pathway.

In clinical treatment, previous studies have shown that SXT is effective in the treatment of cerebral infarction with few adverse reactions. However, caution should be taken against the use with pregnant women and patients with hemorrhagic diseases [173].

### 3.5. Dosage of Natural Medicine in Animals and Humans 

Most of the above natural medicines and their prescriptions can be used in animal experiments and clinics. Therefore, we also reviewed the commonly used medicine doses for the treatment of IS in this section. It can provide a dosage basis for the subsequent application of these natural drugs to IS.

Studies have shown that these drugs are administered in a variety of ways in rats or mice experiments, including oral administration, intravenous injection, intraperitoneal injection, and intramuscular injection. In addition, it has a wide range of administration ranging from 0.27 mg/kg to 100 mg/kg. It is suggested that great attention should be paid to the dose problem when using these drugs to treat IS. However, the clinical use of these drugs is incomplete. The current administration methods used to treat patients with IS are mainly oral and intravenous. It should be noted that the use of TCM prescriptions here is special in China’s clinical practice. For example, in the clinical use of BHD, the quality of each medicine in this prescription is adjusted on the basis of the original according to the patient’s disease. We have summarized the current animal and clinical doses in Table 3.

## 4. Future Prospects

The pathological mechanism of ischemic stroke is complex, which involves the interaction of a variety of molecular signal transduction pathways. The PI3K/Akt signaling pathway is an important pathway in the pathogenesis of IS and plays an important role in anti-neuronal apoptosis, oxidative stress, inflammation, and autophagy. 

Natural medicine, and its prescriptions, can improve and treat ischemic stroke by targeting the PI3K/Akt signaling pathway. Additionally, some progress has been made in its related research. This review systematically expounds the relationship between natural medicine and the PI3K/Akt signaling pathway and the pathogenesis of IS, which is convenient to understand the pathological process of IS as a whole and provide a theoretical basis for the treatment of IS with traditional natural medicine. Although in the current research, natural medicines and prescriptions have been shown to improve and treat IS by targeting the PI3K/Akt signaling pathway, there are still many mechanisms to be explored.

Firstly, the current research on IS and the PI3K/Akt signaling pathway is relatively weak, and some studies have not verified the inhibition or activation of this pathway. Therefore, it is difficult to achieve results using its inhibitors or activators to verify forward and backward mechanisms. In addition, most of the experimental types are animal experiments and the research methods are simple. Clinical verification should be increased, and subsequent animal experiments and clinical observations can be combined to establish a unique database of IS. On this basis, new goals and ideas are put forward. Moreover, the target of PI3K/Akt signaling pathway in the treatment of IS is only supported by classical target proteins, which cannot judge the participation of this pathway in the process of IS as a whole. It needs to be further improved. 

Overall, according to the promotion of the PI3K/Akt signaling pathway in the pathophysiological process of IS, it is suggested that the PI3K/Akt signaling pathway may be an effective way of targeted medicine therapy for IS and it can delay and hinder the occurrence and development of IS. Due to the multi-target therapeutic effect of natural medicine, the specific mechanism of action of related prescriptions and active components through the PI3K/Akt signaling pathway can be explored from the perspective of molecular biology. This can provide new drug development strategies for the improvement and treatment of IS. 

Furthermore, transient ischemic attack (TIA) has also been the focus of clinical research in recent years, and it is closely related to IS. Stroke risk after TIA is highest in the first few days, but it is a benign and reversible cerebral ischemia syndrome. Therefore, early intervention of TIA can also reduce the probability of IS. We can also use TIA as a point to study the relationship between the PI3K/Akt signaling pathway and TIA and its possible mechanism, and discuss how natural medicine can treat TIA and IS through the PI3K/Akt signaling pathway from multiple perspectives and links.

## Figures and Tables

**Figure 1 molecules-27-07963-f001:**
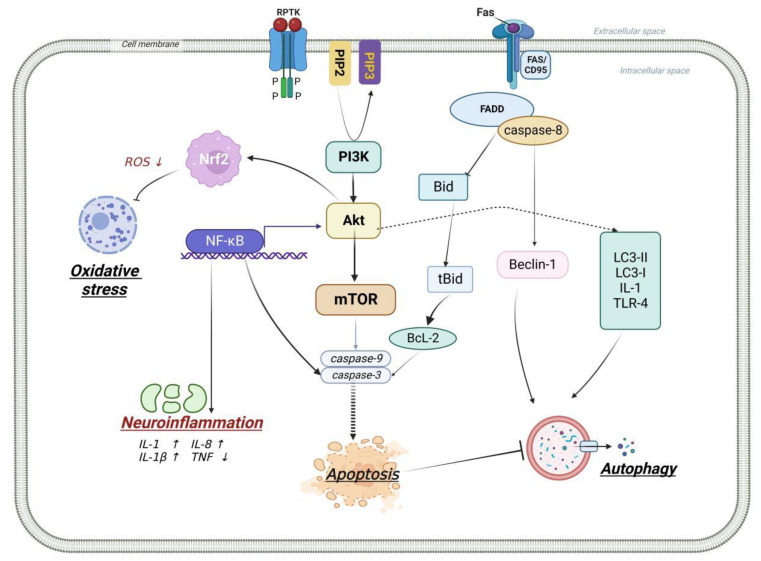
Relationship between PI3K/Akt signaling pathway and apoptosis, inflammation, autophagy, and oxidative stress. After the activation of PI3K/AKT signaling pathway: (1) it can act on the downstream targets such as NF-κB, mammalian target of rapamycin (mTOR), and down-regulate the expression of cysteinyl aspartate specific proteinase-9 (Caspase-9), cysteinyl aspartate specific proteinase-3 (Caspase-3), and apoptosis-related protein B lymphocyte tumor-2 (Bcl-2) associated x protein (Bax), ultimate anti-apoptosis; (2) it up-regulates the expression of low complexity communications codec II/I (LC3-II/LC3-I), Beclin-1, interleukin-1 (IL-1), and toll-like receptor 4 (TLR4) to achieve the effect of anti-autophagy; (3) it inhibits the expression of pro-inflammatory factors induced by NF-κB to achieve the effect of anti-inflammation; (4) activated Akt further activates Nrf2, to achieve the effect of anti-oxidative stress.

**Figure 2 molecules-27-07963-f002:**
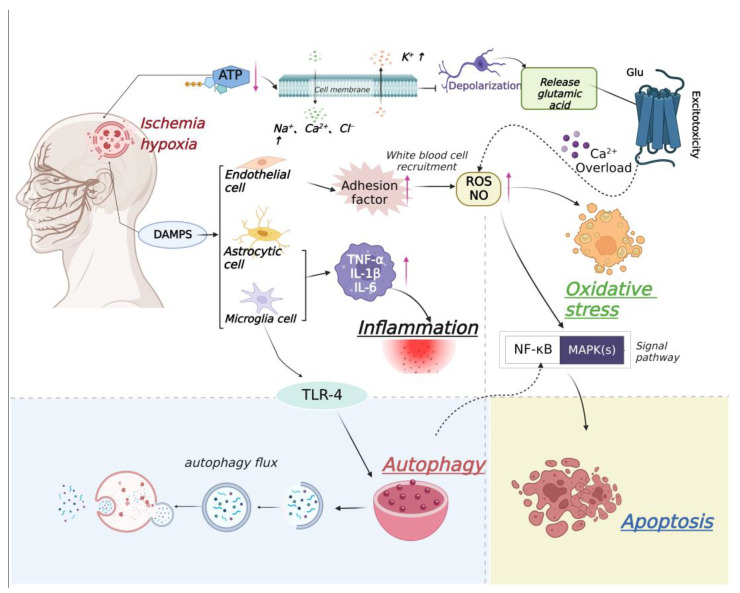
Relationship between ischemic stroke and apoptosis, inflammation, autophagy, and oxidative stress. After cerebral ischemia, the activity of Na^+^/K^+^−ATP enzyme in ischemic penumbra decreases and the imbalance of ion homeostasis leads to cell membrane depolarization and Ca^2+^ influx, resulting in excessive release of glutamate and excitotoxicity. With the influx of a large amount of Ca^2+^, the adhesion factors of endothelial cells increase, which leads to white blood cell recruitment, increases the contents of ROS and nitric oxide (NO), and causes oxidative stress. Then, NF-κB and MAPK signaling pathways are activated, causing cell apoptosis. Microglial and astrocytic cells are also activated, which up-regulate tumor necrosis factor (TNF-α), interleukin-1β (IL-1β), and interleukin-6 (IL-6), finally causing an inflammatory reaction. Meanwhile, microglia cell-mediated down-regulation of Toll-like receptor-4 (TLR-4) induce autophagy.

**Table 1 molecules-27-07963-t001:** Effects of herbal medicine and their active ingredients on PI3K/Akt signaling pathway and their roles in ischemic stroke.

Natural Medicine	Plant Atlas	Active Ingredients	Model	Regulation of PI3K/Akt Signaling Pathway	Main Purposed Effects	Ref(s)
*Herbal* *medicine*						
*Chuanxiong*	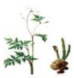	Ligustrazine	PC12 CellsSD ratsHuman amniotic epithelial cells	↑p-PI3K andp-Akt	Anti-neuronal apoptosisAnti-inflammationActivating autophagy	[61,62,63,64,65,66,67]
*Salvia* *miltiorrhiza*	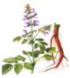	Tanshinone;Danshensu;Salvianolic Acid.	SD ratsNeural stem/precursor cells	↑p-PI3K andp-Akt	Inhibit apoptosisAnti-inflammationAnti-oxidative stress	[68,69,70,71,72,73,74,75]
*Radix* *Angelicae* *sinensis*	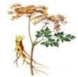	Angelica Polysaccharides;Ferulic Acid;Ligustilide.	PC12 CellsSD ratsCIRI rats	↑p-PI3K andp-Akt	Anti-oxidative stressAnti-neuronal apoptosis	[76,77,78,79,80]
*Astragalus* *membranaceus*	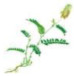	AstragalusPolysaccharides;Astragalus Saponins;Flavonoids.	SD ratsOGD/R HT22 cells	↑p-PI3K andp-Akt	Anti-neuronal apoptosisAnti-inflammation	[81,82,83,84,85,86]
*Safflower*	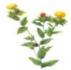	Safflower Yellow	SD rats	↑p-PI3K, p-Akt, GSK3β	Anti-oxidative stressAnti-neuronal apoptosisAnti-inflammation	[87,88,89,90,91,92,93]
*Ginkgo* *biloba leaf*	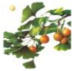	Ginkgo Flavonoids;Ginkgolides.	SD ratsbrain endothelial cell	↑p-Akt,p-GSK3β, VEGF	Anti-inflammationAnti-oxidative stress	[94,95,96,97,98,99]
*Erigeron* *breviscapus*	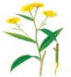	Flavonoids Scutellarin(Breviscapine)	SD ratsA375 cells	↑p-Akt, eNOS	Anti-oxidative stressAnti-neuronal apoptosis	[100,101,102,103]
*Ginseng*	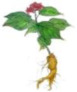	Ginsenoside	MiceSD rats	↑p-Akt,p-mTOR,p-ERK	Anti-oxidative stressAnti-neuronal apoptosisAnti-inflammation	[104,105,106,107,108,109,110,111]
*Radix Paeoniae* *Rubra*	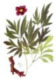	Paeoniflorin	MiceSD rats	↑p-PI3K andp-Akt	Anti-inflammationAnti-neuronal apoptosis	[112,113,114,115]
*Panax* *notoginseng*	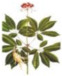	Panax Notoginseng;Saponins.	MiceSD ratsbEnd 3 cellsH9c2 cells	↑p-Akt/Akt,p-mTOR, Nrf2	Anti-neuronal apoptosisAnti-inflammation	[116,117,118,119,120,121]
*Herbal* *prescriptions*						
Buyang Huanwu Decoction		Propyl Gallate;Formononetin;Hydroxysafflor Yellow A;Formononetin; Astragaloside IV;Inosine, Paeoniflorin;Paeonol;Ligustrazine;Ferulic Acid.	HT22 cellsmice	↑p-PI3K, p-Akt, p-Bad	Anti-neuronal apoptosisAnti-inflammation	[122,123,124,125,126]
Taohong Siwudecoction		Hydroxysafflor Yellow A;Paeoniflorin;Paeonol;Ligustrazine; Ferulic Acid.	SD rats	↑p-Akt	Anti-inflammationAnti-neuronal apoptosis	[127,128,129,130]
Xiaoyao San		Saikoside; Ferulic Acid; Ligustilide; Atractylenolide; Paeoniflorin; Albiflorin;Liquiritin; Glycyrrhizic Acid; Pachymic Acid.	PC12 cells	↑p-PI3K andp-Akt	Anti-neuronal apoptosis	[131,132]
Danhong injection		Ferulic Acid;Cryptotanshinone; Quercetin;Anhydrosafflor Yellow B.	SD rats	↑p-PI3K, p-Akt, GSK3β	Anti-inflammationAnti-neuronal apoptosis	[133,134,135,136,137,138]
Sanhua decoction		Flavonoids; Anthraquinones;Coumarins;Phenylpropanoid Glycosides;Alkaloids;Lignans.	SD rats	↑p-PI3K andp-Akt	Anti-inflammationAnti-neuronal apoptosis	[139,140,141]
Xingnaojinginjection		Turmeric;Moschus;Borneolum Syntheticum;Fructus Gardeniae.	SD rats	↑p-Akt, eNOS	Anti-oxidative stressAnti-neuronal apoptosis	[142,143,144,145]

↑ signifies increase/activate

**Table 2 molecules-27-07963-t002:** Effects of insect medicine and their active ingredients on PI3K/Akt signaling pathway and their main purposed effects.

Natural Medicine	Insect Atlas	Active Ingredients	Model	Regulation ofPI3K/Akt Signaling Pathway	Main Purposed Effects	Ref(s)
*Insect* *medicine*						
*Earthworm*	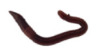	Lumbrokinase	SD ratsmice	↑: p-PI3K andp-Akt	Anti-neuronal apoptosisAnti-inflammation	[147,148,149,150,151]
*Leech*	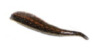	Hirudin; Fibrinolysin.	IPF ratsSD ratsmice	↑: p-Akt	Anti-neuronal apoptosisAnti-inflammation	[152,153,154,155,156]
*Scorpion*	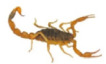	Adenosine;Dipeptides.	SD ratsK562 cell	↑: p-Akt	Anti-neuronal apoptosis	[157,158,159,160]
*Insect* *prescriptions*						
Tongxinluocapsule		Ginsenoside;Hirudin;Denosine;Paeoniflorin.	SD ratsmice	↑: p-Akt	Anti-neuronal apoptosis	[161,162,163,164,165]
Naoxintongcapsule		Adenosine; Hirudin; Tanshinone;Danshensu;Salvianolic Acid.	SD rats	↑: p-Akt	Anti-neuronal apoptosis	[166,167,168,169]
Shuxuetonginjection		Hirudin;Lumbrokinase.	SD rats	↑: p-Akt, VEGF	Anti-oxidative stressAnti-neuronal apoptosis	[170,171,172,173]

↑ signifies increase/activate.

**Table 3 molecules-27-07963-t003:** Doses of active ingredients and related prescriptions of natural medicine.

Medicine	Dose in Animals	Dose in Human	Ref(s)
*Active ingredients*			
Ligustrazine	20 mg/kg/d in rat (i.p.)	80–240 mg/d (i.v.)	[67,174]
Tanshinone IIA	30 mg/kg/d in rat (i.v.) 10 mg/kg/d in mice (i.p.)	60 mg/d (i.v.)	[75,175,176,177]
Ligustilide	20–40 mg/kg/day in rat (p.o.) 5–20 mg/kg/d in mice (i.p.)	/	[178,179]
Hydroxysafflor Yellow A	10–40 mg/kg/d in rat (i.v.) 2 mg/kg/d in mice (i.v.)	25–70 mg/d (i.v.)	[93,180,181]
Ginsenoside Rd	25 mg/kg/d in rat (i.p.) 10–50 mg/kg/d in mice (i.p.)	10–20 mg/d (i.v.)	[182,183,184]
Ginsenoside-Rb1	25–100 mg/kg/d in rat (i.p.)	/	[185]
Ginsenoside-Rg1	20 mg/kg/d in rat (i.p.) 10–40 mg/kg/d in mice (i.p.)	/	[107,186]
Paeoniflorin	40 mg/kg/d in rat (i.p.)	3–9 g/d (i.v.)	[112,187]
Panax notoginseng Saponins	25–100 mg/kg/d in rat (i.p.) 45 mg/kg/d in mice (i.p.)	500 mg/d (i.v.)	[117,188,189]
Salvianolic Acid B	10–20 mg/kg/d in rat (i.p.)	/	[70]
Breviscapus	3–6 mL/kg in rat (i.v.)	/	[190]
Astragaloside IV	2g/kg/d in rat (i.v.) 20 mg/kg in rat (i.p.) 200 mg/kg in mice (i.p.)	/	[82,83,191]
Lumbrokinase	/	1,800,000 units/d (p.o.)	[151]
Hirudin	10–40 mg/kg in mice (p.o.)	2.25 g/d (p.o.)	[153,156]
*Prescriptions*		*Formula/dose*	
Buyang Huanwu Decoction	10–40 g/kg in rat (p.o.) 1.0 g/kg in mice (p.o), twice daily	raw *Astragalus* 30 g, *angelica* 15 g, *longan meat* 15 g, *antler gum* 10 g, *Salvia miltiorrhiza* 10 g, *frankincense* 10 g, *myrrh* 10 g, and *dried pine* 5 g	[192,193,194,195]
Taohong Siwu decoction	4.5–18 g/kg/d in rat (p.o.)	/	[196]
Danhong injection	0.75–3 mL/kg in rat (i.v.), twice daily 3 mL/kg/d in mice (i.m.)	20–40 mL/d (i.v.)	[134,138,197,198]
Sanhua decoction	10 g/kg/d in rat (p.o.)	/	[140]
Xingnaojing injection	0.75–3 mL/kg/d in rat (i.m.) 6 mg/kg/d in mice (i.m.)	20 mL/12 h (i.v.)	[143,199]
Tongxinluo capsule	100 mg/kg/d in rat (p.o.) 0.75–3.0 g/kg/d in mice (p.o.)	1.56–3.12 g/d (p.o.)	[161,162,163]
Naoxintong capsule	0.5 g/kg/d in rat (p.o.)	2.4 g/d (p.o.)	[200,201,202]
Shuxuetong injection	6 mL/kg/d in rat (i.v.) 0.27–1.08 mg/kg/d in rat (i.p.)	6–10 mL/d (i.v.)	[171,172,203]

/ Representative did not report.

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
