# Peer review of "A New Therapeutic Trend: Natural Medicine for Ameliorating Ischemic Stroke via PI3K/Akt Signaling Pathway"

_molecules, 2022, doi:10.3390/molecules27227963_

Round 1

Reviewer 1 Report

The paper by Liu et al. reports the evidence on the effect of natural products on enhancing the PI3K/Akt pathway as an advantageous way to manage the ischemic stroke. This is achieved by firstly briefly introducing the pathology, the various events linking PI3K/Akt and the ischemic stroke (i.e., oxidative stress, apoptosis and autophagy) and then enlisting the natural remedies studied in this field.

Overall, the paper deals with an interesting subject, nicely supported by a robust background before diving into the true review. Nevertheless, the discussion of the remedies fails to be clear and well-written, seeming often quite drafty. Therefore, given the value of the paper, it can be considered for publication after extensive revision of the flaws witnessed.

11) English language must be thoroughly revised, so that punctuation and capital letters after commas or full stops. Authors must revise the many errors found when reading their manuscript.

22) Every acronym must be reported in extenso the first time it appears (i.e., PI3K/Akt, CXCL1, IL, TNF...).

33) The introductions in full of unbound sentences. Authors should make the reading more fluent by better linking the sentences. Moreover, as stated above, many sentences are linked with commas when a full stop must have been used (i.e., lines 60-62).

44) Again, in lines 84-85, Authors must rewrite the confounding sentence.

55) The headings inside all the figures, in my opinion, are not necessary. Therefore, they must be removed.

66) Plant names must be written in full binomial name, in italics and with the family name in capital letter.

77)      In Figure 3, the monographies reported are too small. I suggest Authors to remove them and make a table out of this image to state the plant name and the active ingredients.

88) For each paragraph regarding plant, Authors must remove the word “Definition:” and rewrite the introductory part of each plant to make it more readable.

99) “Animal medicine” and “Treatment of Ischemic Stroke with insect prescription” paragraphs must be removed because the supporting references are from non-impacted journals and very difficult to be found to check what Authors stated. Therefore, the paper would definitely increase in soundness if removing this part.

Author Response

Dear reviewer:

Many thanks for your very useful comments and suggestions to our manuscript. We have modified the manuscript in blue text according to your comments, and the detailed point-to-point responses are listed below: Reviewers' reports:

Point 1. English language must be thoroughly revised, so that punctuation and capital letters after commas or full stops. Authors must revise the many errors found when reading their manuscript.

Response1: Thank you very much for your reminder. We have made a comprehensive revision to the English grammar of the manuscript and the use of punctuation marks.  

Point 2. Every acronym must be reported in extenso the first time it appears (i.e., PI3K/Akt, CXCL1, IL, TNF...).

Response 2: Thank you very much. We have made a detailed supplement to the acronyms that appeared in the manuscript when they first appeared.

Point 3. The introductions in full of unbound sentences. Authors should make the reading more fluent by better linking the sentences. Moreover, as stated above, many sentences are linked with commas when a full stop must have been used (i.e., lines 60-62).

Response 3: Thank you very much for your suggestion. We have rewritten the introduction section.

Point 4. Again, in lines 84-85, Authors must rewrite the confounding sentence.

Response 4: Thank you very much for your suggestion. We have revised the confounding sentence.

Point 5. The headings inside all the figures, in my opinion, are not necessary. Therefore, they must be removed.

Response 5: Thank you for your suggestion. We have removed headings inside all the figures.

Point 6. Plant names must be written in full binomial name, in italics and with the family name in capital letter.

Response 6: Thank you very much for your suggestion. We have regulated the writing of the names of all plants.

Point 7. In Figure 3, the monographies reported are too small. I suggest Authors to remove them and make a table out of this image to state the plant name and the active ingredients.

Response 7: Thank you very much for your suggestion. We have made a table containing the name of the medicine, active ingredient, and effect, etc.

Point 8. For each paragraph regarding plant, Authors must remove the word “Definition:” and rewrite the introductory part of each plant to make it more readable.

Response 8: Thank you very much for your suggestion. It has been revised in the manuscript.

Point 9.“Animal medicine” and “Treatment of Ischemic Stroke with insect prescription” paragraphs must be removed because the supporting references are from non-impacted journals and very difficult to be found to check what Authors stated. Therefore, the paper would definitely increase in soundness if removing this part.

Response 9: Thank you very much for your advice. We are deeply aware of the loophole that the references on “Animal medicine” are from non-impacted journals.However, we think that animal medicine, especially the several medicine mentioned in the manuscript, have great potential in the treatment of ischemic stroke, and can provide more options for the follow-up clinical treatment of ischemic stroke, so we want to retain this part. this is also our original intention to write this review. Therefore,we sorted out the relevant reports again. And we screened the previous references and retained only some of the cutting-edge findings. And then rewrote the "animal medicine" part. At present, the references cited in this section is basically from influential journals.Thank you again for pointing this out so that we can further improve the Manuscript in time.

Reviewer 2 Report

In this review article, authors reviewed the mechanism of different types of natural medicine in the treatment of ischemic stroke through PI3K/Akt signaling pathway, so as to provide a new reference basis for the development of new medicine for ischemic stroke. This is a very interesting review article and could help researchers. I only have some minor comments:

1) All authors focus was drawn toward animal models regarding the effect of these herbal plants, however, ischemic stroke is a fetal disease in humans not more common in animals. Authors could do further research in the databases to look at any studies or even clinical studies for any of these plants or their active ingredients and compare their effect with the current (standard) therapeutic agents.

2) Also, author should add more information on the safety of these herbal plants and their active ingredients.

3) Studies involving a combination of these herbal plant and/or their components should also be addressed and their safety margins.

4) How doses of these herbs and their ingredients could be translated into human? give more details in that regard.

5) During the description of each plant, delete "Definition" and add in brief the methods used for extraction and fractionation.

6) The first word in most sentences stuck to ".", you should add a space.

7) Refernces should be updated as much as you can.

Author Response

Dear reviewer:

Many thanks for your very useful comments and suggestions to our manuscript. We have modified the manuscript in blue text according to your comments, and the detailed point-to-point responses are listed below: Reviewers' reports:

Point 1. All authors focus was drawn toward animal models regarding the effect of these herbal plants, however, ischemic stroke is a fetal disease in humans not more common in animals. Authors could do further research in the databases to look at any studies or even clinical studies for any of these plants or their active ingredients and compare their effect with the current (standard) therapeutic agents.

Response 1: Thank you very much for your suggestion. We have summarized the clinical studies on active ingredients reported in recent years and described them in the manuscript.

Point 2. Also, author should add more information on the safety of these herbal plants and their active ingredients.

Response 2: Thank you very much for your suggestion. We have made a table containing the name of the medicine, active ingredient, and effect, etc. About the point of safety evaluation, we found that not all active ingredients have safety evaluation related reports. Therefore, we tried our best to describe the safety evaluation that has been reported in the corresponding section, with references support. Thank you again for raising this point so that we can improve the manuscript in time.

Point 3. Studies involving a combination of these herbal plant and/or their components should also be addressed and their safety margins.

Response 3: Thank you very much for your suggestion. For the prescriptions mentioned in the manuscript. we also made a table containing the name of the medicine, active ingredient, and effect, etc. About the point of safety evaluation, we also found that not all active ingredients or prescription itself have safety evaluation related reports. Therefore, we tried our best to describe the safety evaluation that has been reported in the corresponding section, with references support. Thank you again for raising this point so that we can improve the manuscript in time.

Point 4. How doses of these herbs and their ingredients could be translated into human? give more details in that regard.

Response 4: Thank you very much for your suggestion. However, the relevant reports we found basically did not mention how the dose of these herbs and their ingredients converted into human. But we think your suggestion is very meaningful, so we summarized the doses of commonly used active ingredients and prescriptions that have been practiced in animals and humans. It is helpful for the follow-up clinic to find the dose basis quickly when using these medicine.

Point 5. During the description of each plant, delete "Definition" and add in brief the methods used for extraction and fractionation.

Response 5: Thank you very much for your suggestion. It has been revised in the manuscript.

Point 6. The first word in most sentences stuck to ".", you should add a space.

Response 6: Thank you very much for your suggestion. It has been revised in the manuscript.

Point 7. Refernces should be updated as much as you can.

Response 6: Thank you very much for your suggestion. We have updated the references.

Author Response

Dear reviewer:

Many thanks for your very useful comments and suggestions to our manuscript. We have modified the manuscript in blue text according to your comments, and the detailed point-to-point responses are listed below: Reviewers' reports:

Point 1. The manuscript is poorly written. In many places of the article the sentence structure is not correct which alters the entire meaning of the sentence. Sometimes the meaning completely obscure or it’s almost impossible for the reader to comprehend what information the authors are trying to convey. For example- the sentence of the lines 48-50 is, “PI3K is an intracellular phosphatidylinositol kinase with serine/threonine (Ser/Thr) kinase and phosphatidylinositol kinase activity, its biological function is rich and its structure is complex” which makes no sense. There are a lot of sentences in the manuscript of this type.

Response 1: Thank you very much for your suggestion.We have made a lot of changes to the language of the manuscript.

Point 2.There is lack of coherence between the titles and the texts under them. For example- in page 4, line 132 the title says, “Neuroinflammation by ischemic stroke associated with PI3k/Akt signaling pathway”- from which one may assume that PI3k/Akt signaling pathway is the underlying mechanism of neuroinflammation; although inside the text it’s the opposite. Additionally, from lines 154-157 it is not clear what message the authors are trying to convey in that sentence. The whole manuscript is full of this type of incoherence.

Response 2: Thank you very much for your suggestion.We have modified the secondary and tertiary headings in the manuscript.

Point 3. In many places some abbreviated terms are used which the readers are exposed to without any prior knowledge. For example- in page 5, line 170 the authors mentioned about tS-ORC without any explanation about what it is. There are many other examples like this inside the text.

Response 3: Thank you very much. We have made a detailed supplement to the acronyms that appeared in the manuscript when they first appeared.

Point 4. In some places the authors used some vague terms like in line 208- ‘in 2h/ model rats’. What does it mean?

Response 4. Thank you very much for your suggestion. It has been revised in the manuscript.

Point 5. Sometimes the topic itself doesn’t make any sense. For example- in page 6, line 240-24, what does ‘Natural medicine for the treatment of Ischemic Stroke through PI3K/Akt signaling pathway’ mean?

Response 5. Thank you very much for your suggestion. and thank you very much for raising this point, so that we can also think deeply about the viewpoint. But we think this topic is meaningful. First of all, natural medicine is the focus of modern medicine research, because many traditional medicine for the treatment of diseases have the disadvantages of narrow treatment window and many side effects. Secondly, the treatment of diseases through the signal pathway is one of the necessary processes to study the medicine mechanism, which can clearly reflect the specific targets of the medicine, and even the relevant factors in the upstream and downstream of the pathway. It can provide new ideas for the study of medicine mechanism.The manuscript summarized the treatment of ischemic stroke by natural medicine through PI3K/Akt signal pathway, which can better provide a mechanism basis for the use of new natural medicine before clinical treatment. To provide reference and new ideas for the basic mechanism research.

Point 6.The authors tried to connect the ischemic stroke with PI3K/Akt pathway but failed to do so because of the lack of proper literature review and lack of explanation

Response 6. Thank you very much for your suggestion. We have supplemented a large number of literatures on ischemic stroke and PI3K/Akt signaling pathway.

Point 7. The figure legends are not explanatory of the figures. For example- in figure 2, it is hard for the readers to follow what the figure actually means. There are several points presented in the legend, but the figure doesn’t follow those points. The authors need to clarify each point inside the figure or alternatively, they can clearly label the figures with steps and then explain those steps in the legend.

Response 7. Thank you very much for your suggestion. We have reinterpreted the figure.

Point 8. In other cases, such as in figure 3, 4 and 5, the legends are presented as the indices of the figures

Response 8. Thank you very much for your suggestion. We have deleted figure 3,  4 and 5, and re-tabulated the contents for readers' convenience.

Point 9. Under the section 3 the authors discussed about the ‘herbal medicine’ and ‘herbal prescriptions’ with the title, “Natural medicine for the treatment of Ischemic Stroke through PI3K/Akt signaling pathway”. They discussed the definitions of each herbal medicines along with their methods of preparation which is completely irrelevant. They hardly showed any evidence of these herbal medicines actually targeting the PI3K/Akt pathway (if not at all). Similar observations apply to ‘herbal prescription’ as well. Even in places where the authors cited the literatures that are related to herbal medicines targeting the PI3K/Akt pathway, they didn’t explain it well.

Response 9. Thank you very much for your suggestion. We have deleted the "definitions" section. And supplemented a large number of literatures on the treatment of ischemic stroke by active ingredients through PI3K/Akt signaling pathway.

Reviewer 4 Report

This manuscript is written exceptionally well. This review manuscript is well-organized with appropriate figures and tables. The use of English language is very good. However, I have a small concern about the manuscript. Based on the authors' personal beliefs and perspectives, they should insert a scheme (in the Prospective section) showing potential mechanisms by which PI3K/Akt signaling pathway is involved in the the pathogenesis of ischemic stroke, TIA, but not hemorrhagic stroke (? maybe). The authors showed a couple of interesting natural compounds/medicine. So, the authors can also add the two interesting natural medicine to the scheme.   

Author Response

Dear reviewer:

Many thanks for your very useful comments and suggestions to our manuscript. We have modified the manuscript in blue text according to your comments, and the detailed point-to-point responses are listed below: Reviewers' reports:

Point 1.This manuscript is written exceptionally well. This review manuscript is well-organized with appropriate figures and tables. The use of English language is very good. However, I have a small concern about the manuscript. Based on the authors' personal beliefs and perspectives, they should insert a scheme (in the Prospective section) showing potential mechanisms by which PI3K/Akt signaling pathway is involved in the the pathogenesis of ischemic stroke, TIA, but not hemorrhagic stroke (? maybe). The authors showed a couple of interesting natural compounds/medicine. So, the authors can also add the two interesting natural medicine to the scheme.   

Response 1: Thank you very much for your suggestion. We have carefully studied the TIA you mentioned. We found that it is a very meaningful research site. And we understand that there is also a certain relationship between TIA and ischemic stroke, which is very helpful to our research. We quite agree with your suggestion. And so far, there are no reports about the relationship between TIA and PI3K/Akt signaling pathway. Therefore, we added the expectation of the relationship between TIA and PI3K/Akt signaling pathway in the prospective section.

Round 2

Reviewer 1 Report

The Authors have almost entirely addressed all the comments highlighted in the first round of revision. Indeed, the paper has gained soundness and robustness by ameliorating references and images, as well as improving fluency and English language. However, I reiterate few concepts that Authors must take into account to render the paper worthy of publication in Molecules.

1)    Botanical names must be written in italics when referring to genus and species. As example, Salvia Miltiorrhiza must be Salvia miltiorrhiza, so all in italics with genus name starting with uppercase letter and species name in lowercase. On this line, family name must not be written in italics but with the starting letter in uppercase and not the whole word (i.e., LABIATAE must be Labiatae). This must be done throughout the manuscript, including tables.

2)    The acronym for ischemic stroke, namely AIS, is fine if preceded by “acute”. Therefore, I suggest Authors to do so in the first line of the abstract and introduction.

3)    The new Table 3 represents an interesting gathering of relevant data, therefore I highly suggests Authors to introduce it in a new paragraph with few introductory lines to highlight the table and catch readers’ attention.

Author Response

Dear reviewer:

Many thanks for your very useful comments and suggestions to our manuscript. We have modified the manuscript in blue text according to your comments, and the detailed point-to-point responses are listed below: Reviewers' reports:

Point 1. Botanical names must be written in italics when referring to genus and species. As example, Salvia Miltiorrhiza must be Salvia miltiorrhiza, so all in italics with genus name starting with uppercase letter and species name in lowercase. On this line, family name must not be written in italics but with the starting letter in uppercase and not the whole word (i.e., LABIATAE must be Labiatae). This must be done throughout the manuscript, including tables.

Response1: Thank you very much for your suggestion. We have modified it according to your request. After these two revisions, we have learned a lot from you. On the issue of botanical names, we will be more rigorous in the future.

Point 2.The acronym for ischemic stroke, namely AIS, is fine if preceded by “acute”. Therefore, I suggest Authors to do so in the first line of the abstract and introduction.

Response2: Thank you very much for your reminder. AIS is acute ischemic stroke, but our theme is ischemic stroke. This is a mistake that should not have occurred, we have changed AIS to IS. And after you pointed out this problem, we have checked the manuscript from beginning to end. And we embellished the language again and revised it word for word. Thanks again for patiently reading our manuscript and list the problem. This is how we can make continuous progress.

Point 3.The new Table 3 represents an interesting gathering of relevant data, therefore I highly suggests Authors to introduce it in a new paragraph with few introductory lines to highlight the table and catch readers’ attention.

Response3: Thank you very much for your suggestion. We have listed a separate paragraph to briefly elaborate on this table.

Reviewer 3 Report

The authors have made significant modifications to the manuscript which improved the quality of the paper a lot. However, there are still a lot of things that needs to be improved. For example- still there are a lot of grammatical mistakes and typos. The sentence structures are poor and sometimes incomplete.

From the scientific point of view, the authors have done a good job to compile the relevant literatures. However, the presentation is poor and sometimes goes beyond the main topic of interest. The authors should focus more on discussing the link between the PI3K/Akt pathway and ischemic stroke rather than elaborating the background information which sometimes make the discussion vague and difficult to follow.

Author Response

Dear reviewer:

Many thanks for your very useful comments and suggestions to our manuscript. We have modified the manuscript in blue text according to your comments, and the detailed point-to-point responses are listed below: Reviewers' reports:

Point 1. The authors have made significant modifications to the manuscript which improved the quality of the paper a lot. However, there are still a lot of things that needs to be improved. For example- still there are a lot of grammatical mistakes and typos. The sentence structures are poor and sometimes incomplete.From the scientific point of view, the authors have done a good job to compile the relevant literatures. However, the presentation is poor and sometimes goes beyond the main topic of interest. The authors should focus more on discussing the link between the PI3K/Akt pathway and ischemic stroke rather than elaborating the background information which sometimes make the discussion vague and difficult to follow.

Response1: Thank you very much for your suggestions after these two reviews. And thank you very much for your affirmation of our last revision. The questions you listed are very intuitive and useful. This also made us aware of the shortcomings of our manuscript. According to your question,we have made the following changes:

1.We have revised the whole manuscript word by word to make sure that each sentence was coherent. And we also re-examined the grammar and punctuation of the whole manuscript and made corrections. In addition, we looked for a professional to embellish the manuscript.

2.We have omitted some background content and highlighted the relationship between medicine, ischemic stroke, and PI3K/Akt signaling pathway. Moreover, we have adjusted the structure of each paragraph to ensure that the main points of the paragraph can be read quickly.

  Dear reviewer, we sincerely appreciate your advice. After these two revisions, we have also learned a lot from you. This made our manuscript a step further. At the same time, these suggestions make us more rigorous. Thank you again for your patience in reading our manuscript.